# Quantitatively assessing early detection strategies for mitigating COVID-19 and future pandemics

Andrew Bo Liu [1,2] ✉, Daniel Lee[2,3], Amogh Prabhav Jalihal[1],
William P. Hanage [4] & Michael Springer [1] ✉

Researchers and policymakers have proposed systems to detect novel pathogens earlier than existing surveillance systems by monitoring samples from hospital patients, wastewater, and air travel, in order to mitigate future pandemics. How much benefit would such systems offer? We developed, empirically validated, and mathematically characterized a quantitative model that simulates disease spread and detection time for any given disease and detection system. We find that hospital monitoring could have detected COVID-19 in Wuhan 0.4 weeks earlier than it was actually discovered, at 2,300 cases (standard error: 76 cases) compared to 3,400 (standard error: 161 cases). Wastewater monitoring would not have accelerated COVID-19 detection in Wuhan, but provides benefit in smaller catchments and for asymptomatic or long-incubation diseases like polio or HIV/AIDS. Air travel monitoring does not accelerate outbreak detection in most scenarios we evaluated. In sum, early detection systems can substantially mitigate some future pandemics, but would not have changed the course of COVID-19.

It has been widely debated which policies, if any, could have mitigated the health impacts of the initial stages of the COVID-19 pandemic in late 2019 and early 2020 as community transmission became established and widespread. Early studies compared non-pharmaceutical interventions (NPIs) such as mobility restrictions[1,2], school closures[3,4], voluntary home quarantine[5] and testing policies[6], and optimized NPI parameters like testing frequency[7], quarantine length[8], testing modality[9], test pooling[10] and intervention timing and ordering[11]. While such NPIs undoubtedly slowed the early spread of COVID-19[12] and previous outbreaks[13,14], there has been little investigation of whether a separate strategy focused on earlier detection of COVID-19 would have enabled more successful mitigation. In theory, earlier detection enables a response when the outbreak is smaller: thus, resource-intensive mitigation strategies like test-trace-isolate become less costly, and the earlier interventions are applied, the larger the number of infections

and deaths that can be delayed until healthcare capacity is increased[15]. However, the relevant question is not whether early-detection helps, but quantitatively, how much of a difference it would make. This question is especially urgent given current international and national policy proposals to invest billions of dollars in such systems[16,17].

Researchers and policymakers have proposed immediate investments in systems to continuously monitor for novel pathogens in (i) patients with infectious symptoms in hospitals and clinics[18,19], (ii) community wastewater treatment plants[20,21], and (iii) airplane sewage or bridge air on international flights[22–24], as well as other sites[25–29]. These three sites have attracted interest because they have been frequent testing sites in COVID-19: hospitals since the pandemic's beginning[30], and wastewater (including wastewater at treatment plants[20], within the sewershed[31], and locally near individual buildings[32]) and air travel more recently[33,34] because hospital cases can lag

[1]Department of Systems Biology, Harvard Medical School, Boston, MA, USA. [2]Department of Biomedical Informatics, Harvard Medical School, Boston, MA, USA. [3]Program in Medical and Population Genetics, Broad Institute of MIT and Harvard, Cambridge, MA, USA. [4]Center for Communicable Disease Dynamics, Department of Epidemiology, Harvard T.H. Chan School of Public Health, Boston, MA, USA. ✉e-mail: andrewliu@g.harvard.edu; michael_springer@hms.harvard.edu

community cases[35]. COVID-19 also spurred methodological innovation and characterization of sampling from these sites[36], particularly wastewater[37–39]. Detecting novel pandemics at these sites has occasionally been piloted[21,40] but has not been implemented at scale, in part because it is unclear if these proposed systems sufficiently expedite detection of outbreaks. The systems under consideration would use multiplex testing for conserved nucleic acid sequences of known pathogen families, exploiting the fact that many past emerging diseases belonged to such families, including SARS-CoV-2 (2019), Ebola (2013), MERS-CoV (2012), and pandemic flu (2009). Proposed technologies include multiplex PCR[41–44], CRISPR-based multiplex diagnostics[45], and metagenomic sequencing[46], possibly implemented with pooling[10].

In this work, to determine whether early detection of novel pathogens at these sites could be effective in changing the course of a pandemic, we first examine whether COVID-19 could have been detected earlier in Wuhan if systems had been in place in advance to monitor hospitals, wastewater or air travel. To do this, we develop, empirically validate, and mathematically characterize a simulation-based model that predicts the number of cases at the time of detection given a detection system and a set of outbreak epidemiological parameters. We then use this model and COVID-19 epidemiological parameters[47] to estimate how early COVID-19 would initially have been detected in Wuhan by the three early-detection systems, and compare this to the actual date of COVID-19 detection. Finally, we use our model to estimate detection times of infectious agents with different epidemiological properties, such as mpox and polio in recent outbreaks[48,49], to inform pathogen-agnostic surveillance for future pandemics.

## Results

### Model to estimate earliness of detection
Previous research[15] and our analysis (Supplementary text, Figs. S1–4 and Table S1) suggest that earlier COVID-19 lockdowns could have delayed cases and deaths. Thus, it is critical to understand which early-detection systems, if any, could have effectively enabled earlier response. To do this, we built a model that simulates outbreak spread and earliness of detection for a given outbreak and detection system (Materials and methods, Supplementary materials). This builds upon branching process models that have previously been used to model the spread of COVID-19[50,51] and other infectious diseases[52]. A traditional branching process model starts from an index case and iteratively simulates each new generation of infections. Our model follows this pattern, but with each new infection, we also simulate whether the infected person is detected by the detection system with some probability (Fig. 1a), and the simulation stops when the number of detected individuals equals the detection threshold and the detection delay has passed. Thus, each detection system is characterized by these three parameters: detection probability, threshold, and delay (Table S2). For example, in hospital monitoring, an infected individual's detection probability is the probability they are sick enough to enter the hospital, which is the hospitalization rate (assuming testing has a negligible false negative rate). In systems that test individuals (hospital and air travel individual monitoring), the threshold is measured in an absolute number of cases. In systems that test wastewater (wastewater monitoring), the threshold is measured in terms of outbreak prevalence because wastewater monitoring can only sample a small percentage of sewage flows, depending on the sampling capacity[53]; thus, a higher number of cases is required to trigger detection in a bigger community (Materials and methods). We gathered literature estimates of detection system and outbreak parameters (Tables S2 and S3) and validated wastewater monitoring sensitivity in independent data (Fig. S5 and Materials and methods, Supplementary materials). We then empirically validated the model by testing its ability to predict the detection times for the first COVID-19 outbreaks in 50 US states in 2020. We gathered the dates of the first COVID-19 case reported by the public

health department of each US state (Table S4) as well as literature estimates of true (tested and untested) statewide COVID-19 case counts in early 2020[54]. Using our model, we were able to predict the number of weeks until travel-based detection in each US state within a mean absolute error of 0.97 weeks (Figs. S6 and S7). To check the robustness of our results, we implemented a second, more complex model with varying reproduction numbers using a Monte Carlo simulation-based package (EpiNow2 v1.3.5[55]). A list of model assumptions can be found in Table S5.

### Early detection's impact on COVID-19 detection in Wuhan
Next, we use our model to examine the detection systems' ability to detect the first major COVID-19 outbreak in Wuhan (Fig. 1b and Table S2). To estimate cases at detection in the actual pandemic, we used literature estimates of total (tested and untested) COVID-19 case counts in Wuhan in late 2019 and early 2020[56]. Our model shows that, on average, hospital monitoring could have detected COVID-19 after 2292 cases (standard error: 76 cases). In reality, the pandemic was identified after 3413 cases on average (standard error: 161 cases). Thus, hospital monitoring would have caught the outbreak 1121 cases earlier (-0.43 weeks earlier), a statistically significant difference with $p = 1.9e$-09 and $t = -6.3$ (df = 141) in a one-sided Welch two-sample $t$ test. Wastewater monitoring would have lagged detection in the actual pandemic; it caught the outbreak after 4,575 cases (standard error: 523 cases), or 1162 cases later, on average ($p = 0.018$; $t = 2.1$; df = 118). We tested this wastewater prediction empirically by calculating the cases until COVID-19 wastewater detection in Massachusetts in early 2020, using literature-estimated Massachusetts COVID-19 cases[54] and Massachusetts wastewater SARS-CoV-2 PCR data[57]; our model prediction was consistent with this analysis (Fig. S8). Because we model wastewater monitoring to detect later in larger communities (Materials and methods, Supplementary materials), the Wuhan result is in part due to Wuhan's 650,000-person catchments. Wastewater monitoring would lead status quo detection of COVID-19 in catchments smaller than 480,000 people, well above the global median catchment size of 30,000 people[58]. Air travel monitoring did not provide any acceleration of detection because of the low probability of simultaneously traveling and being sick.

### Early detection for other diseases: formula and simulation
To make our model easily usable for pathogenic outbreaks beyond COVID-19, we derived a compact formula that approximates the model's simulations. We observed that, without accounting for the delay of $g$ generations between the threshold case's infection and detection, the number of cases until detection, $C$, is a random variable that follows a negative binomial distribution by definition: each infected case is a Bernoulli trial, "success" in that trial occurs when that case enters the detection system (with a probability we name $p_{test}$), and we count the number of cases until the number of successes equals the detection threshold $d$. After accounting for $g$ and the basic reproduction number $R_0$, we derived a formula approximating the mean of $C$ when the outbreak starts in a community covered by the detection system (see Supplementary Text for full derivation):

$$\mathbb{E}[C] \approx \frac{d \times R_0^g}{p_{test}} \qquad (1)$$

We confirmed our formula approximates the simulation model closely by comparing the detection times predicted by both for all the detection systems for multiple diseases (Fig. S9). Thus, the formula allows us to interpret the model and the quantitative relationships between detection times and various variables: the formula shows that the number of cases until detection increases linearly with the detection threshold, increases polynomially with $R_0$ and exponentially with the delay $g$ as $R_0^g$, and decreases as the fraction of cases being tested

a

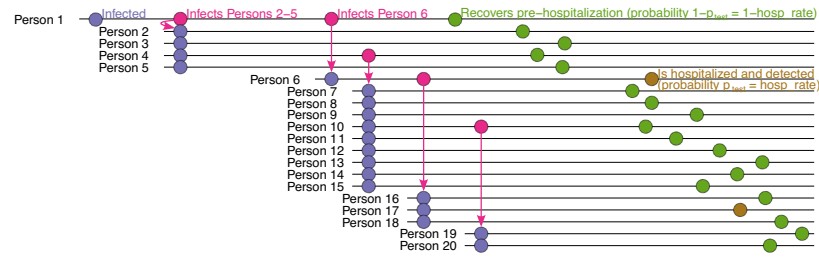

b

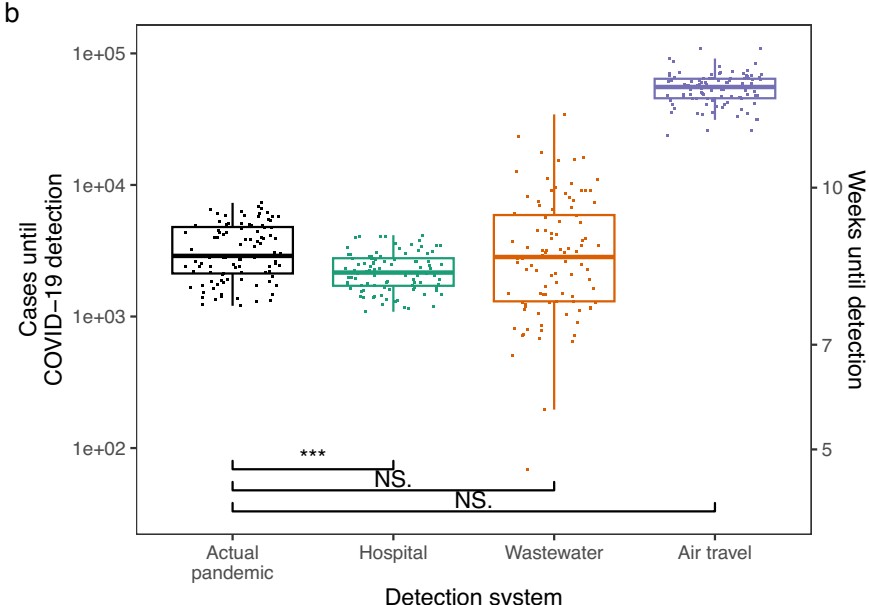

**Fig. 1 | Early detection's impact on COVID-19 detection in Wuhan. a** Schematic of first 20 infections in a simulated run of the detection model. In this run, Person 1 seeds an outbreak in a community covered by a hospital detection system. Each person infects a number of individuals determined by a draw from a negative binomial distribution. Each person is then detected by the detection system with probability $p_{test}$ (gold) or goes undetected (olive); in the hospital system, $p_{test}$ equals the hospitalization rate. **b** Estimated cases until COVID-19 detection in the actual pandemic versus model-simulated cases until detection for proposed detection systems (box plots indicate median (middle line), 25th, 75th percentile (box), and points closest to 1.5× interquartile range (whiskers)). Estimates for the actual pandemic are drawn from ref. 56. Points for proposed detection systems are simulated case counts from the model (actual pandemic (black), hospital (teal), wastewater (orange) and air travel (purple)) assuming a Wuhan-sized catchment (650,000 people). Three, two, and one asterisk(s) signify that the cases upon detection for the detection system are statistically significantly lower than those in the actual pandemic at the 0.001, 0.01, and 0.05 levels, respectively, in one-sided $t$ tests. NS. signifies not statistically significantly lower at $p = 0.05$. $P$ values for systems detecting earlier than in the actual pandemic are 1.9e-09 (hospital), 0.98 (wastewater) and 1 (air travel). Equivalent weeks until detection are shown on the right $y$ axis. Each boxplot shows 100 simulations (points).

increases. This formula also makes the model easily usable for detection systems beyond the ones studied here.

We applied our model to several outbreaks of recent interest—including COVID-19, mpox (2022), polio (2013–2014), Ebola (2013–2016) and flu (2009 pandemic)—and found that the detection systems vary in their success depending on the epidemiological parameters of the agent (Fig. 2, S10 and S11, and table S3). For example, in our model hospital monitoring tends to outperform wastewater monitoring when the hospitalization rate is high, as in the case of Ebola, but tends to underperform for diseases like polio, in which the hospitalization rate is low and when there is high asymptomatic spread in the delay from detection to hospitalization. This is consistent with Eq. (1), as well as previous observations that Ebola was first detected in hospitals[59] and that wastewater monitoring has been more effective than hospital monitoring for detecting polio[60]. Wastewater monitoring performs even better for smaller, 30,000-person catchments (Figs. S12 and S13). We also modeled the status quo detection times for these outbreaks: the number of cases until these outbreaks were detected in

the status quo, without the proposed detection systems in place. We found that early-detection systems can catch outbreaks when they are up to 52% smaller (wastewater for polio) or 110 weeks earlier (hospital for HIV/AIDS) (Figs. S14–S17). Similar results hold for the more complex model: the relative median detection times of the three systems remain the same 97% of the time across the five main diseases (29/30 pairwise comparisons) (Fig. S18).

Because future infectious diseases are likely to have different epidemiological parameters, we generalized the previous analysis and calculated detection times for many possible diseases spanning the epidemiological parameter space (Fig. 3 and S19). As expected, hospital monitoring is the best system for diseases with higher hospitalization rates and lower times to hospitalization. For diseases with higher $R_0$s and times to hospitalization, wastewater monitoring emerges as the best system more often, because hospital monitoring has a longer detection delay (mainly the time from infection to hospitalization) than wastewater (mainly the time from infection to fecal shedding), during which cases grow exponentially with $R_0$. However,

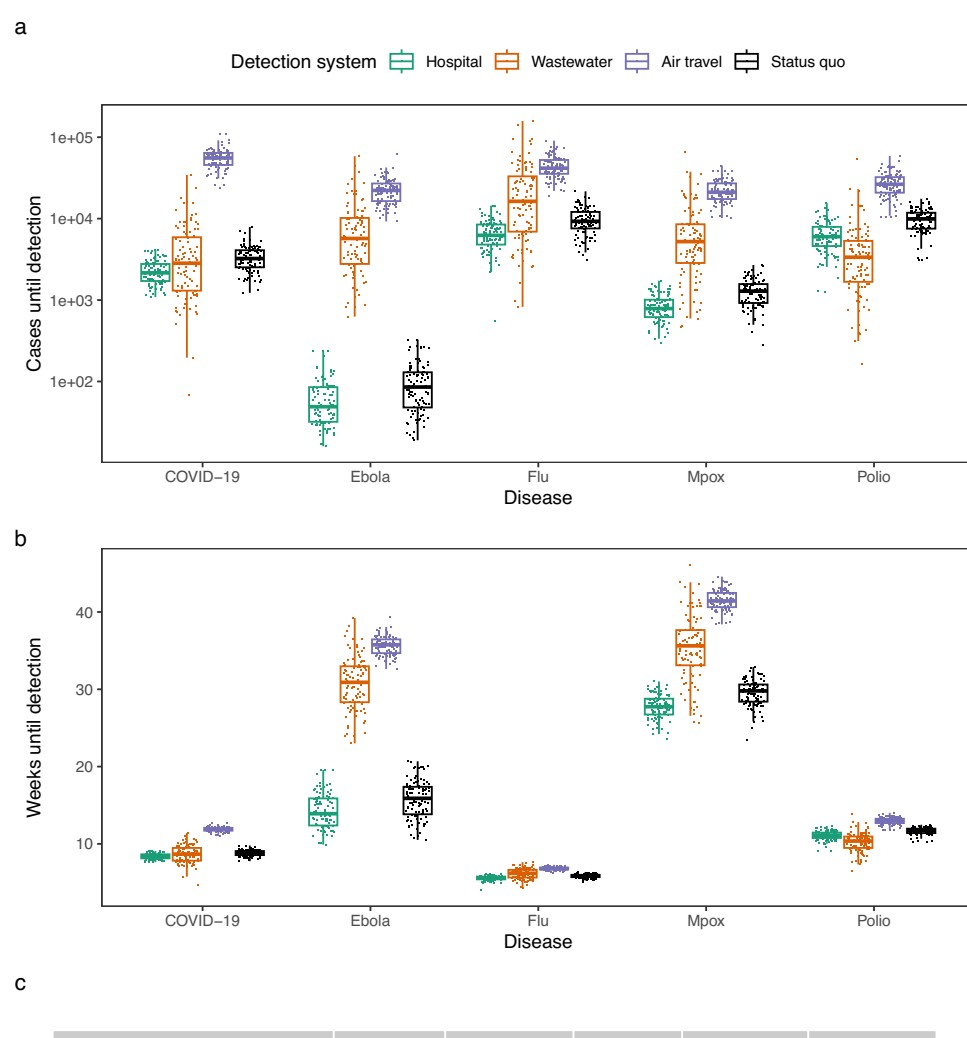

**Fig. 2 | Comparison of detection systems for different diseases. a** Earliness of detection for detection systems in cases across infectious diseases (hospital (teal), wastewater (orange), air travel (purple), and status quo (black)) in a 650,000-person catchment (box plots indicate median (middle line), 25th, 75th percentile (box), and points closest to 1.5× interquartile range (whiskers)). Each boxplot shows 100 simulations (points). **b** Earliness of detection for detection systems in weeks across infectious diseases in a 650,000-person catchment (box plots indicate median (middle line), 25th, 75th percentile (box), and points closest to 1.5× interquartile range (whiskers)). Each boxplot shows 100 simulations (points). **c** Epidemiological parameters of the studied diseases.

| Diseases | COVID–19 | Ebola (2013–2016) | Flu (2009 pandemic) | Mpox (2022) | Polio (2013–2014) |
|---|---|---|---|---|---|
| Hospitalization rate | 0.03 | 0.72 | 0.01 | 0.03 | 0.01 |
| R0 | 2.50 | 1.80 | 1.60 | 1.40 | 1.60 |
| Serial interval (weeks) | 1.00 | 2.10 | 0.30 | 1.40 | 0.60 |
| Time to hospitalization (weeks) | 1.50 | 2.30 | 0.50 | 2.00 | 1.60 |
| Probability of fecal shedding | 0.50 | 0.50 | 0.25 | 0.50 | 0.90 |
| Dispersion | 0.70 | 0.10 | 0.10 | 0.10 | 0.10 |

this holds mainly for diseases with high probability of fecal shedding and low hospitalization rate. Air travel monitoring, which did not perform well in the previously modeled diseases (Figs. 1 and 2), actually performed best for a few diseases for which fecal shedding is low (disadvantaging wastewater monitoring) and the time to hospitalization and $R_0$ are too large (disadvantaging hospital monitoring).

## Discussion

Our results show that the benefits of early-detection systems vary from marginal (0.4 weeks earlier for COVID-19) to significant (110 weeks earlier for HIV/AIDS) (Figs. 1B, 2, and S17). Our detection time model (Fig. 1a) can be used for many diseases and detection systems, including other systems beyond this study[25,26], by varying the fraction

of the infected population being tested in each system. Some further points are worth emphasizing. First, early-detection only aids mitigation if it leads to a coordinated early response. Many factors beyond detection affect the pace of response, including the economic and political feasibility of lockdowns, the availability of medicines and personal protective equipment, and whether there are pre-determined policies to be implemented upon detection. Second, when deciding to invest in these systems, one must consider factors such as cost-effectiveness and whether the system provides evidence of disease severity. Although wastewater monitoring gives earlier detection than hospital monitoring in multiple diseases (Fig. 3a), it does not discriminate between mild and severe disease (although sequencing could detect lineages known to cause severe illness). In contrast,

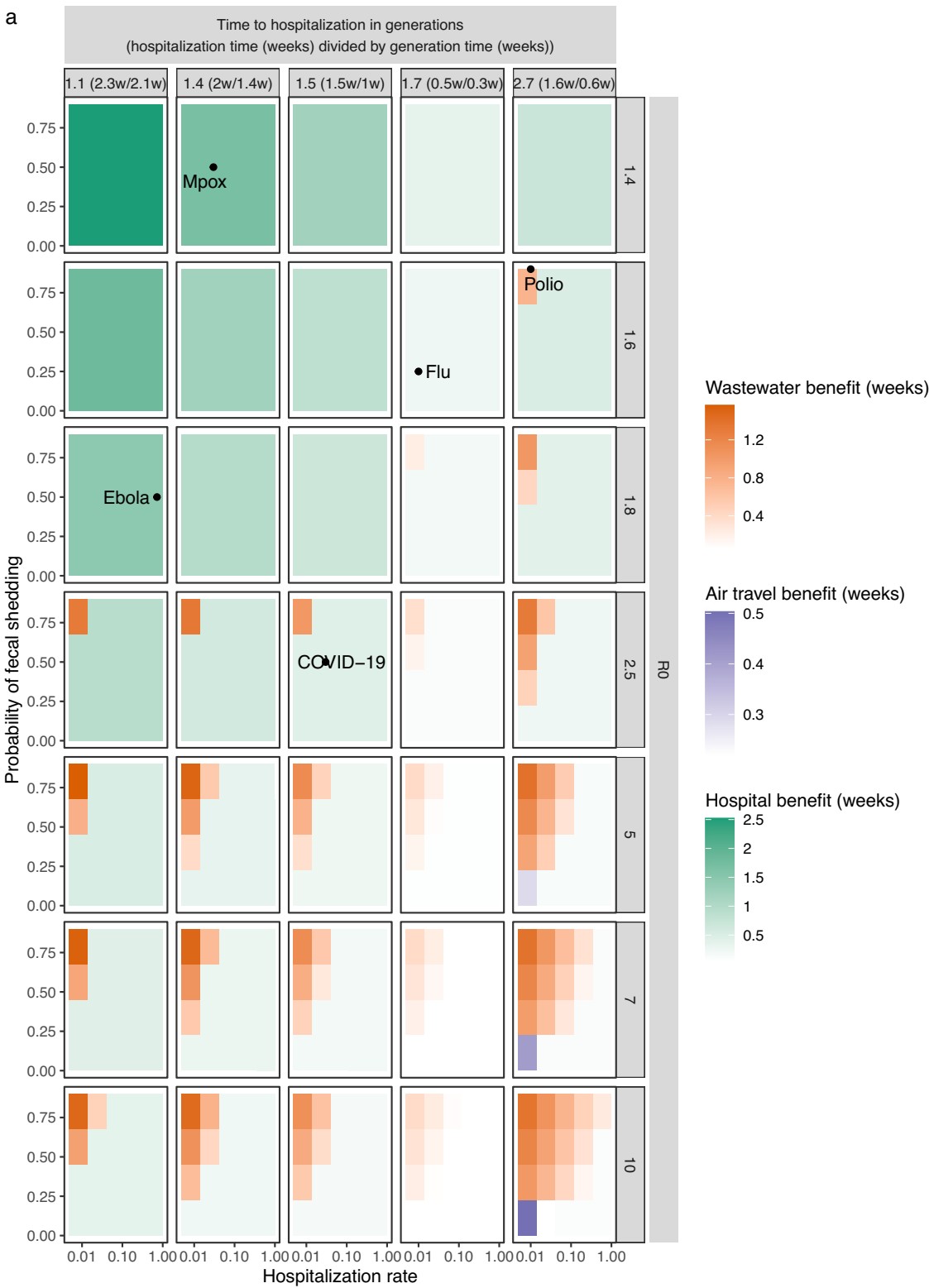

**Fig. 3 | Comparison of detection systems across the space of possible diseases of varying epidemiological parameters. a** Average weeks gained over status quo detection by the proposed detection systems across the epidemiological space of possible diseases. Within each panel, each uniformly colored cell corresponds to a specific disease with the hospitalization rate and probability of fecal shedding indicated on the x and y axes, as well as the $R_0$ and time to hospitalization (generations) indicated by the panel row and column. The cell has a hue corresponding to the detection system that detects the disease the earliest (hospital (teal), wastewater (orange) and air travel (purple)) and an intensity corresponding to the number of weeks gained by the earliest system over status quo detection. Times are calculated by the derived mathematical approximation in a 650,000-person catchment.

hospital monitoring provides evidence that the detected pathogen produces symptoms that require hospital treatment. Third, our model is meant to be used now, in advance of future pandemics, and not in the early months of a novel pandemic, because early-detection systems must be set up in advance of the next pandemic to be effective. Because we do not know the epidemiological parameters of the next pandemic, our study assesses how these systems would perform for a wide, representative variety of diseases with different epidemiological parameters, in order to quantify these systems' benefits in general.

These results can inform ongoing international and national policy debates about which policies are needed to mitigate future pandemics. In the wake of COVID-19, the World Health Organization Intergovernmental Negotiating Body is actively negotiating a new treaty on international pandemic preparedness which updates the International Health Regulations (2005). Drafts of this treaty highlight "early warning and alert systems" as key measures[16]. Similarly, the presidential administration of the United States has proposed investing $5.3 billion over 7 to 10 years in early warning and real-time monitoring systems, including in hospitals and wastewater[17]. In this study, we have assessed detection systems' detection times and have developed a model to assess current and future detection system proposals. Along with additional cost-effectiveness analysis and technical pilots[21], these results can help inform which detection systems are most effective and thus worth funding in pandemic preparedness efforts.

## Methods
### Description of model predicting cases at detection
Our branching process-based model predicts the cumulative number of cases at the time of detection for a given detection system and outbreak. It follows the approach of branching process simulation models used previously to model the spread of COVID-19[50,51] and other infectious diseases[52], but with the main added step of simulating each infected person's chance of being detected by the detection system. The values for parameters for detection systems can be found in Table S2. The values for epidemiological parameters for outbreaks ($R_0$, serial interval, dispersion, hospitalization rate, and time to hospitalization) can be found in Fig. 2 and Table S3. As in previous models[52], we assume the offspring distribution (the number of secondary cases infected by each primary case) is negative binomial with mean $R_0$.

We generally follow past detection system proposals[19,21] to determine the implementation details of each system in our model. Our model assumes the following. In hospital monitoring, hospitals would test for high-priority pathogen families (e.g., coronaviruses) in patients presenting with severe infectious symptoms in hospital emergency departments[19]. Similarly, in wastewater monitoring, governments would test for pathogens in city wastewater treatment plants daily, and monitor for high and increasing levels of high-priority pathogen families[21]. In air travel monitoring, we model testing of individual symptomatic passengers (differs from proposals to monitor airplane sewage[22] or bridge air) on incoming international flights for the same pathogens. The parameters of these systems are shown in Table S2.

Our model also accounts for different delays involved in different detection systems. For example, if the 500th case of a COVID-19-like outbreak triggers the detection threshold in both the hospital and wastewater monitoring systems, because of the significant delay from infection to hospitalization compared to the delay from infection to fecal shedding, the wastewater system would catch the outbreak earlier.

In systems that test individuals (hospital and air travel individual monitoring), the threshold is measured in an absolute *number* of cases. In systems that test wastewater (community and air travel wastewater monitoring), the threshold or sensitivity is measured in prevalence (cases as a *percentage* of the population)[53,61,62]. To predict the number of cases and time to detection, we need to convert this percentage back to a number of cases, so the wastewater detection time depends on the catchment population size.

To estimate wastewater sensitivity measured in prevalence, we used data from[53]. This study conducted PCR testing for SARS-CoV-2 1687 longitudinal wastewater samples from 353 sampling locations in 40 US states in early 2020, and synced these with publicly reported local daily new COVID-19 case counts. This enables us to estimate a distribution of the wastewater sensitivity: the lowest case count required to trigger positive detection in wastewater. Of the 353 sampling locations, 47 had both SARS-CoV-2-positive and negative samples such that local case counts on days of positive samples were all higher than those on days of negative samples. We thus knew each sampling location's sensitivity is between the maximum of case counts on negative sample days and the minimum of case counts on positive sample days. We took the midpoint of this maximum and minimum as the location's sensitivity; this gave us 47 local sensitivities. We fitted this to a log-normal distribution, yielding a median of 2.5 daily new cases per 100,000 people. As expected, this distribution is similarly shaped but slightly left-shifted from the distribution in Fig. 2b of ref. 53 (median 3.7 per 100,000), because the latter distribution is an upper bound of the former.

To use this distribution in our model, in each simulation run, we first randomly drew a wastewater sensitivity from this distribution, and then we needed to convert this *reported incidence i* to the *true (reported and unreported) number* of cases shedding fecally into public wastewater systems up to the time of wastewater detection. We converted as follows. Let day T be the day on which the incidence *i* is reported. First, we assumed the wastewater SARS-CoV-2 level on day T is proportional to the number of COVID-19 cases who are fecally shedding on day T, which we estimate as the number of fecal shedders infected 2 days before, given the dominant peak in fecal shedding on day 2 of infection[61]. We infer the number of fecal shedders infected on day T-2 from the incidence as follows. To account for underreporting, we first estimate a true daily incidence of 5.7× *i* with symptom onset on day T, based on estimates of the ratio of true (dated by symptom onset) to reported (dated by reporting date) COVID-19 cases in the United States in early 2020[54]. (This study's abstract reports true cases are 5–50× reported ones, but this refers to the early March 1–April 4, 2020, period. We calculated the factor of 5.7 from the study's data when we use the fuller March 1-May 16 period, which overlaps better with the February-June 2020 period in[53] and reflects less underreporting as the pandemic developed and testing capacity increased. We calculated this underreporting factor as an average of state-level underreporting factors, weighted by frequency of each state among the wastewater samples in ref. 53.) Finally, we multiply by (a) the fraction of cases who shed fecally (0.5[63]) and (b) the fraction of people connected to central sewage (0.8 in the US[64], which is the area from which the[53] threshold is derived). This gives us the one-time prevalence of cases *p* who contribute to the wastewater SARS-CoV-2 level on day T. For a given catchment with population *c*, this one-time number of cases is *cp*, and we estimate the cumulative number of fecal shedders up to this time as $\sum_{t=T}^{0} \frac{cp}{R_0^{t/7}} \approx \int_{t=0}^{T} R_0^{t/7} dt$, where $T = \log_{R_0^{1/7}}(cp)$ is the number of days for the daily exponential outbreak incidence curve to grow from 1 to *cp* cases.

To check this estimate, we identified studies that compared wastewater and hospital COVID-19 trends[20,53] found that trends in wastewater SARS-CoV-2 values led trends in hospital admissions by 1-4 days in New Haven (catchment size 2e + 05). We estimate that wastewater detection would lead hospital detection of COVID-19 in New Haven by −0.8 to 3 weeks (90% CI). This is consistent with the 1-4d lead estimate from[20]. Similarly[53], found that trends in wastewater led those in clinical data by 4 days in Massachusetts (catchment size 2,300,000). Their clinical data are dated by date of reporting rather than sample gathering; assuming that hospital admissions are 5 days ahead of tests by date of reporting[20], then wastewater is 5d-4d = 1 day behind hospital admissions. We estimate that wastewater detection would lead

hospital detection of COVID-19 in Massachusetts by −4 to −0.09 weeks (90% CI). This is consistent with the 1-day lag estimate from[53].

## Validation of model in US states

We gathered two sources of data for each state: dates of COVID-19 detection and COVID-19 case counts in early 2020. For the former, we searched media reports and US state public health press releases to determine the dates of the first COVID-19 case reported in each US state. Sources for each state's detection date are listed in Table S4. We were able to identify such dates for all 50 states.

For the latter, we used literature estimates of true (tested and untested) COVID-19 case counts, which incorporate COVID-19 mortality data to deal with variation in testing capacity among states[54]. We received a time series of weekly symptomatic COVID-19 case estimates for March 1-May 10, 2020 and divided by a symptomatic rate of 0.55 to get an estimate of total (symptomatic and asymptomatic) cases[47]. We specifically used estimates from the adjusted mMAP (mortality maximum a posteriori) method because[54] had mMAP estimates for all 50 states, whereas other methods from the same study were missing estimates for various states. We fit an exponential curve of case counts in each state to extrapolate cases back to January 2020. In the data we received, all states had case data for all weeks from March 1–May 10, 2020.

We used our model to predict the weeks until detection in each US state ($y$ axis in Fig. S6). Because most US states detected their first case by travel (Table S4), we modeled a travel-based detection system similarly to how we modeled the aforementioned detection systems. We simulated a growing stream of imported travel cases ($R_0^i$ cases for the $i$th generation and global $R_0 = 2.5$), and as for the other detection systems, we simulated infection and detection steps for each generation, except that we only allowed travel-associated cases to be detected. We assumed that the state COVID-19 outbreaks had the same values for all epidemiological parameters except for $R_0$, which we allowed to vary by state to account for state-specific conditions. We obtained state-specific $R_0$ values from ref. 65. The values for shared parameters were obtained from literature (Table S3). We used a detection delay of 12 days (5-day incubation period[47] plus 7-day test and reporting turnaround in early 2020 in the US[66]) because many first cases were detected following symptoms. The only parameter we were unable to precisely estimate from literature was the probability of a travel case being detected. We noted that this rate was at most the COVID-19 symptomatic rate (0.55[47]) and at least the hospitalization rate (0.03[47]): in the highest-detecting scenario, every symptomatic case would volunteer to be tested; in the lowest-detecting scenario, only hospitalized travel cases would get flagged for testing. So we chose a rate of 0.1, near the two rates' geometric mean. The predicted detection time for each state (the $y$-value reported in Fig. S6) was the mean of 100 simulations.

We compared these predictions to ground truth estimates in each state ($x$ axis in Fig. S6). These ground truth estimates were calculated by summing the aforementioned weekly case counts from the first week of January 2020 until the date of detection in that state (Fig. S7).

## Early detection's impact on COVID-19 detection in Wuhan

We used our model to examine whether the early-detection systems could have detected COVID-19 earlier than in the actual pandemic. To do this, we used two data sources: (1) literature estimates of total (tested and untested) COVID-19 case counts in late 2019 and early 2020[56] and (2) simulation output from our model. We then used (1) to calculate the cumulative number of cases when COVID-19 was actually detected, and compared this to results from (2).

For (1), we chose to use estimates from[56], which quantifies both the time of SARS-CoV-2 introduction into humans and the time series of cases following said introduction. These estimates are based on phylodynamic rooting methods applied to SARS-CoV-2 sequence data, combined with epidemic simulations and accounting for

epidemiological data on the first known cases of COVID-19. These estimates improve upon previous attempts to time SARS-CoV-2's introduction into humans, which are solely based on phylodynamic rooting methods to quantify the time to the most recent common ancestor of SARS-CoV-2 sequences[67].

As instructed by[56], we utilized 'BEAST.-primary.IH.Dec10_16.linB.Dec15_25.linA.cumulativeInfections.timedGE-MF_combined.stats.pickle' from GitHub[68] to obtain the distribution of daily case counts. Based on the fact that there were at least six COVID-19-related hospitalizations by 2019-12-29[69], we narrowed the distribution to those epidemic simulations with the top 25 percent of hospitalizations and case counts. We simulated 100 draws from this distribution, and then took the number of cases on 2019-12-29 in each simulation to get 100 values for the distribution of cumulative cases at detection in the actual pandemic ('Actual pandemic' boxplot in Fig. 1B). We chose 2019-12-29 as the date that COVID-19 was detected in the actual pandemic, because this was the date of the first report of an outbreak of pneumonia cases to health authorities in Wuhan[70].

For (2), we ran our model for COVID-19 (see Table S3 for the epidemiological parameters used) and all three detection systems (100 simulations for each system). For each detection system, this gave us the estimated number of cases until the detection of COVID-19 if that system had been in place at the start of the pandemic. We assumed the system was present in the community in which COVID-19 originated. We compared each system to the actual pandemic, and determined that detection could have occurred earlier with the system if there was a statistically significant difference in cases until detection between the actual pandemic and the simulated world with the system (Fig. 1b). Statistical significance was assessed by a 1-sided $t$ test in which the alternative hypothesis was that the detection system performed better.

We could empirically test our model predictions for the cases until wastewater detection by using literature-estimated total COVID-19 cases in Massachusetts[54] and Massachusetts wastewater SARS-CoV-2 data[57] in early 2020. We aimed to use these to estimate the cases until COVID-19 wastewater detection in Massachusetts in early 2020, but because Massachusetts wastewater sampling for COVID-19 started only after the Massachusetts outbreak was underway, wastewater samples were positive for SARS-CoV-2 on the first day of testing, so this first day of testing was later than when wastewater detection could have caught SARS-CoV-2 if wastewater detection had been in place in advance. Thus, we could only calculate an upper bound on the true cases until detection. We utilized the wastewater time series from the Massachusetts Water Resources Authority (MWRA) website and synced it with the COVID-19 case count time series (Fig. S8). We multiplied the Massachusetts statewide cases by 0.33 (equal to 2,300,000/6,900,000) because the MWRA data covers 2,300,000 people, out of 6,900,000 people in Massachusetts in 2020. We then summed these case counts up to the date of apparent wastewater detection to get an upper bound for cases at detection, and checked whether our model prediction was consistent with this bound.

## Simulated versus mathematically approximate detection times

We compared the model simulations of cases until detection with our derived mathematical formula, Eq. (1) (Fig. S9). The points in Fig. S9 are the same as in Fig. 2a. The dashed lines are generated by plugging values into Eq. (1) for each detection system: we plugged in the detection threshold, detection probability, outbreak $R_0$, and detection delay (measured in number of generations, i.e., serial intervals) for $d$, $p_{test}$, $R_0$, and $g$, respectively.

## Comparison of detection systems for different diseases

We applied our model to several outbreaks of recent interest: COVID-19, mpox (2022), polio (2013–2014), Ebola (2013–2016) and flu (2009 pandemic) (Fig. 2a). Because of the lack of data on the number of cases

at the time of detection in previous outbreaks (except for the COVID-19 data used in Fig. 1b), we used our model to estimate status quo detection times for the outbreaks. Because many recent outbreaks have been detected in healthcare settings[59,69,71,72], we assumed status quo detection was similar to hospital monitoring, except with a lower detection probability per case ($p_{test}$) to reflect that symptomatic cases are less likely to be tested for a panel of diseases without the proposed systematic, proactive testing scheme. The per-case detection probability for status quo was set to 0.67 times that of hospital monitoring to match our modeled status quo detection times for COVID-19 with those estimated independently by ref. 56 (Fig. 1b).

## Software

Analyses and figures were generated by code at https://github.com/abliu/early-detection/releases as well as tidyverse (v1.3.1).

## Reporting summary

Further information on research design is available in the Nature Portfolio Reporting Summary linked to this article.

## Data availability

Detection time data are available at https://github.com/abliu/early-detection/releases. Estimated COVID-19 case counts in late 2019 and early 2020 are available from Pekar et al.[56] (https://github.com/sars-cov-2-origins/multi-introduction), and the US wastewater threshold data are from Wu et al.[53] (https://www.sciencedirect.com/science/article/pii/S0043135421005984?via%3Dihub#sec0019). In the supplementary analyses, the Massachusetts Water Resources Authority wastewater data are from https://www.mwra.com/biobot/biobotdata.htm, national COVID-19 case counts in early 2020 are from the Johns Hopkins Center for Systems Science and Engineering 2023 (https://github.com/CSSEGISandData/COVID-19), and US state COVID-19 case counts in early 2020 are from Lu et al.[54] (https://journals.plos.org/ploscompbiol/article?id=10.1371/journal.pcbi.1008994).

## Code availability

Code is available at https://github.com/abliu/early-detection/releases[73].

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

## Acknowledgements

We gratefully acknowledge Mauricio Santillana, Nicholas B. Link, Fred S. Lu, and Andre T. Nguyen for sharing their estimates on COVID-19

incidence in the US states. We also acknowledge Jonathan Pekar, Joel Wertheim, and Michael Worobey for sharing their estimates on COVID-19 incidence in late 2019 and early 2020. We acknowledge Michael McLaren, Becky Ward, and Quincey Justman for feedback on the manuscript. We thank the following for their financial support: Lynch Foundation Fellows Program in Systems Biology at Harvard Medical School (A.B.L.), National Library of Medicine grant T15LM007092 (D.L.), National Institutes of Health grant R01GM120122 (A.P.J.), CDC contract 200-2016-91779 (W.P.H.), and National Institutes of Health grant R01GM120122 (M.S.). This publication was also made possible by the New England Pathogen Genomics Center of Excellence (Cooperative Agreement NU50CK000629). This project has been funded (in part) by contract 200-2016-91779 with the Centers for Disease Control and Prevention. Disclaimer: The findings, conclusions, and views expressed are those of the author(s) and do not necessarily represent the official position of the Centers for Disease Control and Prevention (CDC).

## Author contributions

A.B.L. and M.S. conceived of the study. A.B.L., A.P.J., and M.S. developed the methodology. A.B.L. wrote the software and validated the results. A.B.L. and D.L. performed the mathematical analysis. A.B.L. gathered data on epidemiological parameters, wastewater sensitivities, case counts, and dates of detection. A.B.L. and M.S. wrote the original draft. A.B.L., D.L., A.P.J., W.P.H., and M.S. reviewed and edited the manuscript. A.B.L. and M.S. developed the figures. A.B.L., W.P.H. and M.S. supervised the study. M.S. acquired funding for the study.

## Competing interests

W.P.H. is a member of the scientific advisory board and has stock options in BioBot Analytics. M.S. is a cofounder of Rhinostics and consults for the diagnostic consulting company Vectis Solutions LLC. The other authors declare that they have no competing interests.
