## [Peer Review File · Nature Communications]

Quantitatively assessing early detection strategies for mitigating COVID-19 and future pandemicsEditorial Note: This manuscript has been previously reviewed at another journal that is not operating a transparent peer review scheme. This document only contains reviewer comments and rebuttal letters for versions considered at *Nature Communications*.

REVIEWERS' COMMENTS

Reviewer #2 (Remarks to the Author):

I apologise to the reviewers for the delay in this report. I also thank them for the time in addressing my concerns.

1) The comparison with EpiNow2 is reassuring and does address my concerns.

2) I do not find the CDC toilet example convincing, and it still reinforces my concerns, simulating and fitting is one thing, but in this example having a causal explanation for a policy seems like a large leap. The extensive discussion on travel bans highlighted how challenge air monitoring is.

3) Yes, ok agreed, the more advanced mathematical models are not needed here.

4) I am sorry but on the epi parameters I still remain unconvinced, I feel you have introduced an empirical set up to estimate things, and then make an argument that you would "would perform for a wide, representative variety of diseases with different epidemiological parameters". I still fail to see how if I had this model in February 2020 it would give me any new insights.

5) With regards to waste water monitoring stochasticity, what i mean is there will be random variance in estimates, but more importantly there will be systematic bias between sites, this has consistently been a challenge when using waste water data.

Reviewer #3 (Remarks to the Author):

Thank you for providing a rigorous response to my comments and concerns. The manuscript does merit publication in its revised form as it provides a useful approach to assessing surveillance strategies for a range of pathogenic targets. I believe that this could and will be improved as more data and knowledge is accrued across the developing surveillance landscape, but the current manuscript is novel and valuable for acceptance at this point in time.

Response to reviewer final comments:

Reviewer #2

I apologise to the reviewers for the delay in this report. I also thank them for the time in addressing my concerns.

We thank Reviewer #2 for taking the time to read our response and provide additional feedback.

1) The comparison with EpiNow2 is reassuring and does address my concerns.

We thank Reviewer #2 for suggesting this direction.

2) I do not find the CDC toilet example convincing, and it still reinforces my concerns, simulating and fitting is one thing, but in this example having a causal explanation for a policy seems like a large leap. The extensive discussion on travel bans highlighted how challenge air monitoring is.

We mentioned the CDC proposal as an example of an early detection scheme that has been pursued without an analysis of its benefits in terms of how much response time would be gained. We agree that our estimates are simulated and fitted; this is the main feasible approach given the lack of empirical data on early detection times. We are not using our estimates to claim that travel bans did or did not work during COVID-19. The debate over travel bans (e.g. <https://www.sciencedirect.com/science/article/pii/S1326020023005289>, or <https://pubmed.ncbi.nlm.nih.gov/32031668/> for a pre-COVID review) mainly evaluates travel bans based on their ability to keep a disease outside of a country or slow its spread; our paper is focused on how air travel monitoring might help with **the time of detection**. These are separate questions. To clarify this, we have edited the abstract by replacing “Monitoring of air travel provides little benefit in most scenarios we evaluated” with “Air travel monitoring does not accelerate outbreak detection in most scenarios we evaluated” (lines 22-23 in latest version).

3) Yes, ok agreed, the more advanced mathematical models are not needed here.

4) I am sorry but on the epi parameters I still remain unconvinced, I feel you have introduced an empirical set up to estimate things, and then make an argument that you would "would perform for a wide, representative variety of diseases with different epidemiological parameters". I still fail to see how if I had this model in February 2020 it would give me any new insights.

To clarify, this model is meant to be used **in advance** of future pandemics (e.g. February 2019 rather than February 2020 to build off of your example). We don't know the parameters of the next pandemic, so we use epidemiological parameters spanning a variety of historical diseases to estimate the systems' performance for diseases **in general**. For example, we can conclude from Figure 2 that wastewater PCR monitoring in large catchment sizes tends to detect later than hospital or status quo monitoring for many diseases, but detects earlier than both for diseases with high fecal shedding rates (e.g. polio) or in smaller catchments. This indicates that, going forward, a mixed strategy of hospital and wastewater monitoring would be beneficial in large catchments because each can catch a different class of disease. We have clarified this on lines 228-233 (“Third, our model is meant to be used now... in order to quantify these systems' benefits in general”).

5) With regards to waste water monitoring stochasticity, what i mean is there will be random variance in estimates, but more importantly there will be systematic bias between sites, this has consistently been a challenge when using waste water data.

We agree that variance and bias exist; to address this, we used a dataset of wastewater sensitivity thresholds from 1687 longitudinal samples from 353 different sites (see lines 276-277).

Reviewer #3

Thank you for providing a rigorous response to my comments and concerns. The manuscript does merit publication in its revised form as it provides a useful approach to assessing surveillance strategies for a range of pathogenic targets. I believe that this could and will be improved as more data and knowledge is accrued across the developing surveillance landscape, but the current manuscript is novel and valuable for acceptance at this point in time.

We thank Reviewer #3 for taking the time to read our response and provide additional feedback.